# Single-cell profiling of kinase substrate phosphorylation by single-molecule imaging

Takuya Hidaka [1,2¤], Ryotaro Motoya[2,3☯], Gao Jintian[2,4☯], Sooyeon Kim[1,2,5], Yuichi Taniguchi [1,2,4,5,6]*

1 RIKEN Center for Biosystems Dynamics, Suita, Osaka, Japan, 2 Institute for Integrated Cell-Material Science (iCeMS), Kyoto University, Sakyo-ku, Kyoto, Japan, 3 Graduate School of Science, Kyoto University, Sakyo-ku, Kyoto, Japan, 4 Graduate School of Biostudies, Kyoto University, Sakyo-ku, Kyoto, Japan, 5 Graduate School of Pharmaceutical Sciences, The University of Tokyo, Bunkyo-ku, Tokyo, Japan, 6 Graduate School of Frontier Biosciences, Osaka University, Suita, Osaka, Japan

☯ These authors contributed equally to this work.
¤ Current address: Institute of Molecular Biotechnology of the Austrian Academy of Sciences (IMBA), Vienna BioCenter (VBC), Vienna, Austria
* taniguchi@mol.f.u-tokyo.ac.jp

## Abstract

Protein phosphorylation regulates diverse cellular processes, yet its analysis at the single-cell level remains challenging due to the low abundance of phosphoproteins. Here, we present a highly sensitive system for profiling phosphorylation of kinase substrates in individual cells. The method integrates fluorescence labeling of single-cell proteomes, immunoprecipitation using antibodies recognizing phosphory-lation within specific amino acid motifs, miniaturized SDS-PAGE, and single-molecule detection using a custom-built light-sheet fluorescence microscope. We applied this approach to analyze substrates of casein kinase 2 (CK2) in HeLa cells treated with the phosphatase inhibitor calyculin A. Bulk and pseudo-single-cell analyses confirmed treatment-induced accumulation of phosphorylated CK2 substrates and demonstrated quantitative performance over biologically relevant input ranges. Importantly, true single-cell measurements revealed heterogeneous phosphorylation patterns across molecular weight regions, highlighting cell-to-cell variability in CK2 signaling that is obscured in bulk analyses. This platform enables profiling of the phosphorylation states of a wide range of kinase substrates in individual cells and provides a foundation for dissecting heterogeneous signaling dynamics.

## Introduction

Protein phosphorylation is one of the most prevalent post-translational modifications, regulating a wide range of cellular processes, including signal transduction, cell division, metabolism, and apoptosis [1]. Mediated by protein kinases and reversed by phosphatases, phosphorylation typically modulates the activity, stability, local-ization, or interactions of target proteins. Dysregulation of phosphorylation signaling

**Data availability statement:** All relevant data are within the paper and its Supporting information files.

**Funding:** This work was supported by Japan Society for the Promotion of Science (JSPS) [22K14786 (Grant-in-Aid for Early-Career Scientists) and 22KJ3130 (Grant-in-Aid for JSPS Fellows) to T. H., 20H00460 (Grants-in-aid for Scientific Research (A)), 19H05545 and 20K20458 (Grants-in-aid for Challenging Pioneering Research) to Y. T., 19K15718 and 22K14800 (Grants-in-aid for Early-Career Scientists) to S. K.], Japan Science and Technology Agency [JPMJPR25J4 (PRESTO) and JPMJAX1914 (ACT-X) to S. K. and JPMJCR2334 (CREST) to Y. T.], Suntory Foundation for Life Science [SunRiSE] to Y. T., and The Asahi Glass Foundation [Continuation Grants for Young Researchers] to Y. T. The funders had no role in study design, data collection and analysis, decision to publish, or preparation of the manuscript.

**Competing interests:** The authors have declared that no competing interests exist.

cascades has been associated with a range of diseases, including cancer and neuro-degenerative disorders, highlighting their clinical relevance [2,3].

Mass spectrometry (MS)-based proteomics has become the gold standard for large-scale identification and quantification of proteins and their post-translational modifications, including phosphorylation. Enrichment strategies such as immobilized metal affinity chromatography (IMAC), metal oxide affinity chromatography (MOAC) and immunoprecipitation have improved the sensitivity of phosphopeptide detection [4]. However, these bulk analyses typically require millions of cells, averaging signals across heterogeneous populations and masking cell-to-cell variability.

Single-cell phosphoproteomic analyses revealed that rapid therapeutic resistance can arise from rapid, heterogeneous rewiring of signaling networks rather than changes in average protein levels [5]. This underscores the clinical importance of resolving cell-to-cell signaling heterogeneity to identify actionable resistance mechanisms that are masked in bulk analyses. Although recent advances in single-cell mass spectrometry have enabled the detection of hundreds to thousands of proteins in individual cells using miniaturized sample processing and ultrasensitive MS instrumentation [6], applying these approaches to single-cell phosphoproteomics remains a significant challenge, since the abundance of phosphoproteins is quite low, exacerbates the already demanding sensitivity requirements. Antibody-based methods such as single-cell Western blotting and the single-cell barcode chip (SCBC) can detect specific proteins and their modifications in single cells but are limited in measuring only a handful of proteins [5,7].

In this study, we present a novel system that enables profiling of the phosphorylation status of specific kinase substrates in single cells. The method integrates three key components: (1) fluorescence labeling of proteins from single cells and immunoprecipitation using antibodies targeting phosphorylation in a specific amino acid motif, (2) SDS-PAGE using miniaturized polyacrylamide gel to resolve the captured proteins, and (3) high-sensitivity single-molecule imaging using a custom-built light-sheet fluorescence microscope to count individual protein molecules within the gel. This approach enables quantitative analysis of phosphorylation of a wide range of kinase substrates in individual cells.

## Results

### Single-cell phosphoprotein profiling system based on single-molecule imaging

The abundance of proteins in human cells spans an extremely wide dynamic range of approximately seven orders of magnitude. While a small number of highly abundant proteins account for core cellular functions, proteins involved in regulatory processes such as signal transduction are generally present at much lower abundance [8]. Moreover, only a subset of these proteins are phosphorylated at any given time and signaling-relevant phosphoproteins constitute only a small fraction of the total proteome, which further complicates single-cell analyses of phosphorylated proteins [9]. To address this challenge, we employed a custom-built light-sheet fluorescence microscope, the Planar Illumination Microscope for Single Molecule Imaging for All Purpose (PISA) [10,11].

Unlike traditional light-sheet microscopes that require specialized sample chambers or immersion geometries, the PISA microscope offers an open-top design resembling standard inverted biological microscopes [10,11]. This design simplifies sample handling and supports continuous and rapid scanning over large areas and depths (up to 200 µm). By confining laser excitation to thin optical planes and using a highly sensitive camera, our setup of PISA microscope achieves single-molecule counting and can quantify fluorescently labelled protein molecules down to attomolar or zeptomolar concentrations in gel-based assays such as SDS-PAGE.

Based on this highly sensitive approach, we recently developed a single-cell proteome analysis method termed single-cell PAGE-PISA [12]. The workflow comprises isolation of single cells, lysis and fluorescent labelling of proteins with NHS ester dye, SDS-PAGE to separate proteins by molecular weight, and volumetric imaging of the gel using PISA to detect individual fluorescently-labelled protein molecules. Single-cell PAGE-PISA quantifies protein abundances down to ~$10^3$ copies per band (~300 attograms), outperforming traditional staining-based SDS-PAGE by 4–6 orders of magnitude in sensitivity. This allows profiling of approximately $10^5$ protein copies per single mammalian cell. The system also supports pseudotime and cell-type classification analysis from obtained protein expression profiles.

To expand this system for profiling phosphorylated proteins, we combined single-cell PAGE-PISA with single-cell immunoprecipitation using antibodies that recognize phosphorylation within specific amino acid motifs (Fig 1a). Following single-cell sorting, cells were lysed and cellular proteins were fluorescently labeled using an NHS-ester dye. The lysate was then incubated with antibody-conjugated magnetic beads to capture proteins phosphorylated at amino acid motifs recognized by the target kinase. After immunoprecipitation, the beads were washed with an $MgCl_2$ solution. Following removal of the $MgCl_2$ solution, the bead pellets were incubated in a gellan gum solution on a magnetic stand. Because gellan gum gelation is induced by magnesium ions retained within the bead pellets, hydrogel formation occurred only around each pellet. As a result, each single-cell sample formed a bead pellet embedded within a hydrogel matrix. This sample preparation confines the low abundant phosphoproteins in micro-meter sized space which increases local concentration of analyte and enables efficient detection. The pellets were loaded onto a custom-fabricated thin (~100 µm, 2 cm in length; S1 Fig) polyacrylamide gel that was covalently attached to a glass slide using a custom magnetic stand (S2 Fig) to make sure the pellets settle on the top of the gel properly, and the captured phosphoproteins were then resolved by electrophoresis using a 3D-printed chamber (Fig 1b).

After the electrophoresis, the gel was scanned using a PISA microscope for single-molecule imaging. To scan the gel efficiently, the imaging was performed in two steps using a PISA microscope controlled by custom-made program (Fig 1c). First, bright field imaging of the gel top is performed to locate the bead pellets. After that, each lane defined by the bead pellets, was scanned for fluorescence imaging. Compared with conventional polyacrylamide gels, the small size of the bead pellets together with the thin gel ensure that most resolved protein molecules remain within the field of view during scanning. Fluorescent images obtained from each single-cell sample are projected into a single 2D image (S4a and S4b Fig), and the final phosphoprotein expression pattern was generated by plotting the number of detected foci against migration distance (Figs 1d and S4a).

## Profiling phosphorylated CK2 kinase substrates in bulk and single chemically treated cells

To evaluate the performance of our system, we analyzed the effects of chemical treatment on the phosphorylation of CK2 kinase substrates at both bulk-cell and single-cell levels. CK2 is one of the earliest identified Ser/Thr kinases and phosphorylates motifs containing multiple acidic residues, with a consensus sequence of pS/pT-D/E-X-D/E, where the +3 position is most critical, followed by the +1 position [13]. CK2 supports a wide range of cellular functions and is implicated in cancers, human infections, neurological diseases, autoimmune disorders, and other pathological conditions, and it is considered a clinically important target [14].

HeLa cells were treated with calyculin A, an inhibitor of protein phosphatases 1 and 2A, which induces the accumulation of phosphoproteins. First, protein phosphorylation at the bulk-cell level was evaluated by western blotting using an

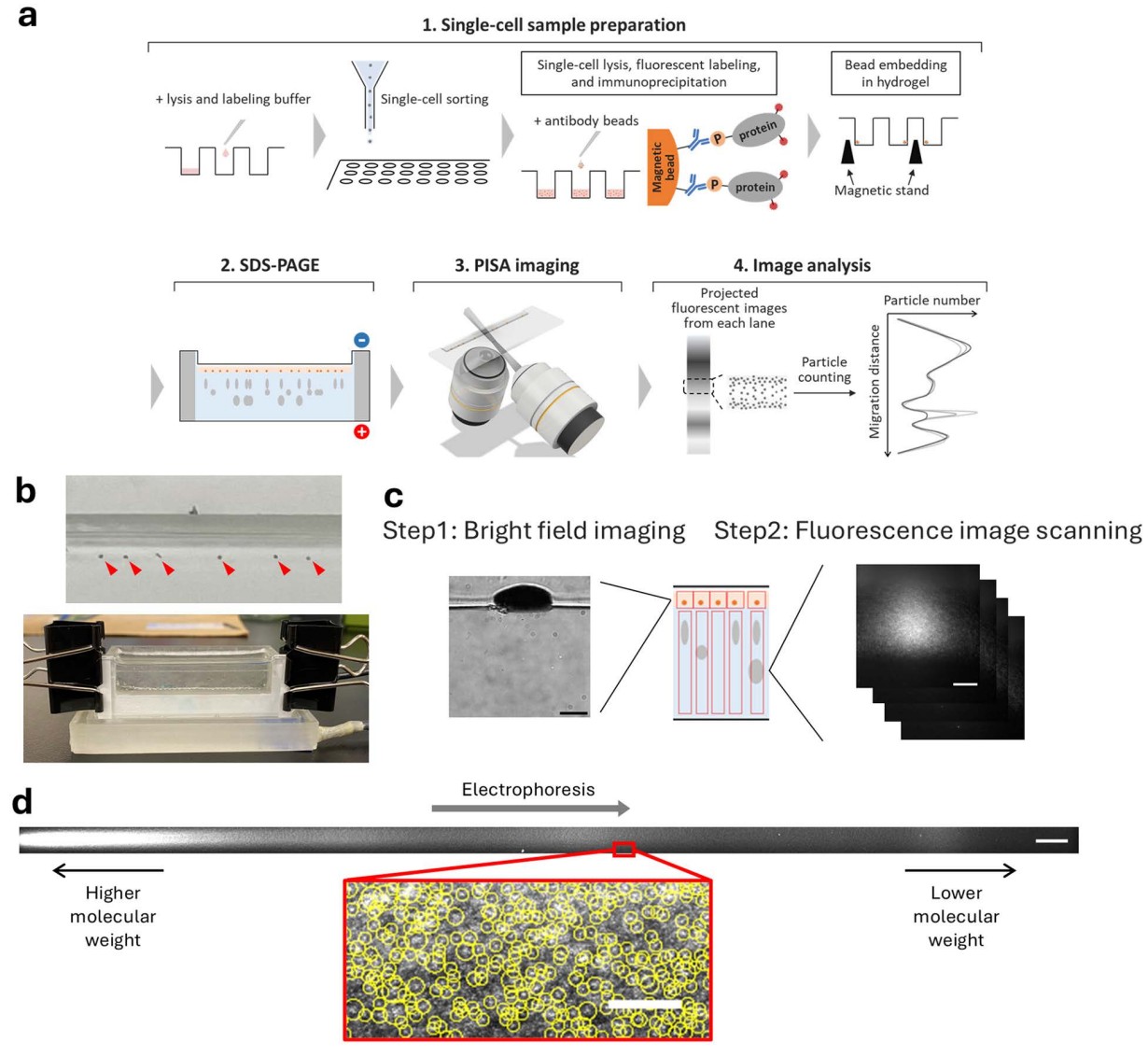

**Fig 1. Overview of the single-cell profiling system for phosphorylated kinase substrates. a)** Schematic overview of the single-cell phosphoprotein profiling system developed in this study. **b)** Bead pellets loaded onto a polyacrylamide gel (indicated by red arrows) and the miniaturized gel electrophoresis system. **c)** Representative images of bead pellets acquired by initial bright-field imaging and subsequent fluorescence image scanning. Scale bar, 80 μm. **d)** Electrophoresis pattern obtained by projection of fluorescence images acquired using PISA microscope scanning. The projected images were subjected to particle detection, and detected foci are indicated by yellow circles in the example image. Scale bars, 400 μm (whole pattern) and 4 μm (zoomed panel).

antibody that recognizes the phosphorylated (pS/pT)DXE motif present in a subset of CK2 substrates (Fig 2a). It should be noted that proteins detected with the anti-(pS/pT)DXE antibody are referred to as phosphorylated CK2 substrates in this study, although other kinases can occasionally phosphorylate the same motif. As expected, calyculin A treatment resulted in stronger signals, particularly in the high–molecular weight region, indicating accumulation of phosphorylated proteins.

To examine whether this change in phosphorylation was also reflected in immunoprecipitated proteins, we performed immunoprecipitation after fluorescent labeling of proteins in bulk cell lysates and resolved the precipitated proteins by

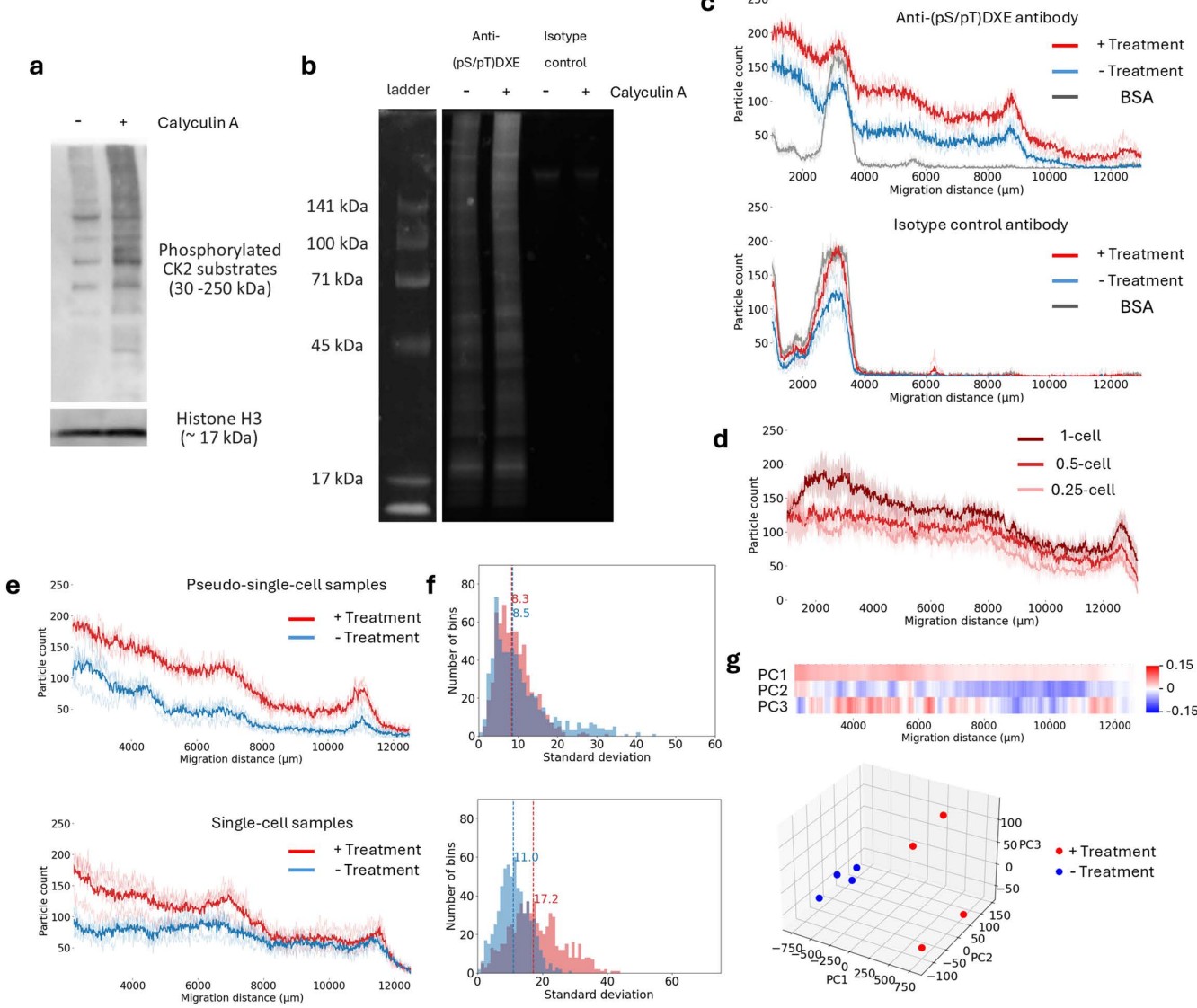

**Fig 2. Calyculin A–induced changes in protein phosphorylation. a)** Western blot analysis of bulk HeLa cells treated with calyculin A using an anti-(pS/pT)DXE antibody and anti-histone H3 antibody as loading control. The uncropped blot image is shown in S3a Fig. **b)** Immunoprecipitation of phosphorylated CK2 substrates from bulk HeLa cell lysates. Proteins were fluorescently labeled with an NHS-ester dye prior to immunoprecipitation, and the precipitated proteins were resolved by SDS–PAGE. The uncropped gel image is shown in S3b Fig. **c)** Phosphoprotein profiles of proteins pulled down from homogenized pseudo-single-cell samples using an anti-(pS/pT)DXE antibody or an isotype control antibody, obtained with the developed system. Thick lines indicate the mean, and thin lines represent individual samples (n = 3). Representative projection images and their overlays with detected particles are provided in S1 File. **d)** Phosphorylation profiles of CK2 substrates from homogenized calyculin A–treated samples diluted to the equivalent of 1-, 0.5-, and 0.25-cell levels (n = 3, mean ± SD). **e)** Comparison of CK2 substrate phosphorylation profiles between homogenized pseudo-single-cell samples and single-cell samples. Thick lines indicate the mean, and thin lines represent individual samples (n = 4). **f)** Histograms of the standard deviation at each bin, calculated from the phosphorylation profiles in (e). Broken lines indicate the median values. **g)** Principal component analysis (PCA) of CK2 substrate phosphorylation profiles from the single-cell samples shown in e. The PCA score plot (bottom) and a heatmap showing the weights of different migration positions to each principal component (top) are shown.

SDS–PAGE (Fig 2b). Consistent with the western blot results, more proteins were pulled down in the treated samples. This increase was not observed when an isotype control antibody was used, indicating that the observed difference was not due to nonspecific binding to antibody beads.

Next, we investigated whether these phosphorylation changes could be detected at the single-cell level. Bulk cell lysates were diluted to single-cell equivalents, and these "pseudo-single-cell" samples were immunoprecipitated using the same anti-(pS/pT)DXE antibody or an isotype control antibody, followed by analysis using our phosphoprotein profiling system. CK2 substrate phosphorylation profiles were obtained by detecting particles in projected images and binning the number of detected foci according to migration distance, which correlates with molecular weight.

In agreement with the bulk-cell analyses, calyculin A–treated samples exhibited profiles that were clearly distinct from those of untreated samples, showing increased phosphoprotein signals (Fig 2c). Because the pseudo-single-cell samples were derived from homogenized bulk lysates, similar profiles were obtained among samples prepared under the same conditions, supporting the robustness of the system. Importantly, samples immunoprecipitated with isotype control antibodies showed almost no signal except for a peak at a migration distance of 3500 μm corresponding to full-length antibody, and a small peak around 6000 μm corresponding to fragmented antibody. This indicates that the signals detected with the anti-(pS/pT)DXE antibody were not caused by nonspecific binding to antibody beads. Furthermore, immunoprecipitation of bovine serum albumin yielded almost no signal, confirming that the detected particles in cell lysate samples were derived predominantly from cellular proteins. Because the strong full-length antibody signal interferes with quantification of sample-derived signals in that region, only profiles beyond the full-length antibody peak were used for subsequent analyses.

To evaluate the quantitative performance of the system, lysates from bulk calyculin A–treated cells were further diluted to 0.25- and 0.5-cell equivalents. A moderate correlation between dilution factor and the number of detected foci was observed across a broad molecular weight range (Fig 2d), supporting the quantitative nature of our particle-based assessment of protein abundance.

Finally, to investigate cell-to-cell heterogeneity in CK2 kinase substrate phosphorylation during calyculin A treatment, we applied the system to single-cell samples both with and without homogenization (Fig 2e). As described above, homogenized samples diluted to the single-cell level (pseudo-single-cell samples) were used to assess technical variability, as they are expected to exhibit identical phosphoprotein profiles and thus reflect only experimental fluctuations. Both pseudo-single-cell and true single-cell samples exhibited similar overall patterns under the same treatment conditions, with broad peaks around 4500, 6500, and 11000 μm. However, compared with pseudo-single-cell samples, true single-cell samples showed greater variability in their electrophoretic profiles among replicates, especially after calyculin A treatment suggesting heterogeneous cellular responses. Consistently, single-cell samples exhibited higher standard deviations in particle counts at each bin (Fig 2f), indicating that the observed variability exceeds technical noise and reflects cell-to-cell heterogeneity in phosphorylation states.

To systematically identify regions contributing to this heterogeneity, we performed principal component analysis (PCA) of the phosphorylated CK2 substrate profiles (Fig 2g). In PCA plots, samples with similar profiles are positioned close to each other. Consistent with this principle, untreated samples, which exhibited low variability, clustered tightly, whereas treated samples with higher variability were more widely dispersed. By visualizing the weights of each migration region to individual principal components as heatmaps, we identified regions driving variation along each PC axis. PC1 showed positive contributions from nearly all regions, suggesting that it captures the overall phosphorylation level of CK2 substrates. Additionally, calyculin A–treated samples are separated into two distinct clusters along the PC3 axis. Bands observed at migration distances of approximately 4,000, 6,500, and 11,500 μm exhibit strong positive loadings on PC3, indicating that CK2 substrates within these regions contribute disproportionately to cell-to-cell heterogeneity in phosphorylation under calyculin A treatment.

Taken together, these results demonstrate that our system enables quantitative profiling of phosphorylated kinase substrates at the single-cell level and that the resulting profiles can be used to analyze cell-to-cell heterogeneity in protein phosphorylation and to identify phosphoprotein bands that contribute to this variability.

## Discussion

In this paper, we introduced a system to profile phosphorylation of substrates of specific kinase in single cells by counting effectively the number of fluorescently labelled and immunoprecipitated phosphoproteins resolved in confined space of polyacrylamide gel mostly covered in the field-of-view of the PISA microscope. We demonstrate that our system can capture the change in CK2 kinase substrate phosphorylation by Calyculin A treatment at single-cell level and systematically identify protein bands causing cellular variation in protein phosphorylation. By targeting phosphorylation in specific amino acid motifs and resolving proteins in relatively long distance (~2 cm), the system can include a wide range of kinase substrates without prior knowledge of specific target proteins, which is usually required in antibody-based systems such as antibody array [5] and single-cell Western blotting system [7].

However, it should be noted that there are several limitations in the current system. First, the current workflow involves multiple experimental steps, among which protein separation using a very thin (~100 μm) miniaturized polyacrylamide gel is particularly critical. The quality of gel preparation strongly affects electrophoretic mobility, separation efficiency, and overall reproducibility between experiments. We anticipate that reproducibility and resolution can be further improved by implementing more stringent control of gel preparation conditions, including temperature, humidity, and deoxygenation, as well as by optimizing gel composition. We also demonstrated that single-cell samples exhibit greater variability in phosphoprotein profiles than pseudo-single-cell samples, which supports the system's ability to capture cell-to-cell heterogeneity beyond technical noise alone (Fig 2e and 2f). However, a more rigorous assessment of inter-experimental variability is needed to distinguish biological heterogeneity from technical variation more reliably. This would involve increasing replicate numbers for both pseudo-single-cell and single-cell groups across independent experiments and establishing statistical metrics for counts at each region.

Second, since the efficiency of fluorescence labeling with NHS-ester dye depends on the number and accessibility of amine groups in lysin residues, proteins with fewer accessible lysines may be labeled less efficiently. Consequently, labeling efficiency is not uniform across proteins and in general, larger or more basic proteins tend to be labeled more efficiently than smaller or more acidic proteins. Therefore, comparisons are most reliable within the same molecular weight region across different samples, rather than between distinct regions.

Third, since the system relies on immunoprecipitation using an antibody against a specific amino acid motif, only substrates containing the target motif are assessed, and substrates of other kinases sharing the same or similar motifs might be included in the assay. Therefore, orthogonal validation is required to definitively assign the detected proteins as substrates of target kinases.

The resolution power of the current system is limited, and the detected regions are expected to contain multiple protein species. Increasing the migration distance or focusing on a narrower region using a modified gel concentration could improve the resolution of protein separation. Variability in labeling efficiency depending on lysine accessibility might also provide further contrast between protein species within the same region.

Nevertheless, even with these improvements, the system would only allow us to narrow down candidate phosphoproteins in regions of interest based on estimated molecular weight, and protein identification would still not be achievable without further technical advancements. This represents the primary limitation of the current system and such improvements are essential for the biological interpretation of phosphoprotein profiles and an important direction for future development.

The combination of 1D/2D gel electrophoresis and mass spectrometry (GE-MS or 2DGE-MS) has been used intensively with bulk cell samples to identify novel factors [15,16], and this powerful approach can be extended to single cell

samples by combining our system with mass spectrometry. The required development is to transfer proteins in specific regions to mass analysis. This might be achieved by using a photo-degradable polyacrylamide gel [17] and inducing gel degradation on microscope stage with a light sheet different from illumination light for single-molecule imaging to collect protein molecules in region of interest for subsequent mass analysis. Actually, modern mass spectrometers are in principle sensitive enough to detect even a few molecular copies, but signals from low-abundance proteins are often masked by highly abundant species in single-cell mass spectrometry due to limited dynamic range and ion suppression [18]. By coupling the single-cell phosphoprotein profiling system reported here with targeted mass spectrometric analysis of protein bands that drive cell-to-cell heterogeneity, it would become possible to uncover novel kinase substrates critical for signaling dynamics that have previously been masked by highly abundant proteins. This framework establishes a foundation for future studies of heterogeneous signaling dynamics at the single-cell level.

## Materials and methods

### Chemicals and materials

General chemicals and materials were obtained from standard commercial suppliers and used without further purification.

### Cell culture

HeLa cells were purchased from JCRB Cell Bank (Japan, JCRB9004). The cells were maintained in Dulbecco's Modified Eagle Medium (Thermo Fisher Scientific, 10566–016) supplemented with 10% fetal bovine serum (FBS) (Corning, 35–079-CV) in a humidified $CO_2$ incubator at 37°C.

### Bulk cell lysate preparation

The cells were collected after trypsinization and $1 \times 10^6$ cells were resuspended in a fresh medium supplemented with 100 nM Calyculin A (Sigma, 208851). The cells were incubated in a humidified $CO_2$ incubator at 37°C for 30 minutes. After the incubation, the cells were washed with PBS twice and lysed in 100 μL of RIPA buffer (20 mM HEPES (pH7.5), 0.05% NP-40, 1 mM EDTA, 150 mM NaCl, 1x Phosphatase Inhibitor Cocktail (Nacalai Tesque, 07575−51) and 1x Protease Inhibitor Cocktail (Nacalai Tesque, 25955−24)). Lysates were clarified by centrifugation at $12,000 \times g$ for 10 min at 4°C. The protein concentration was determined with Protein Assay BCA Kit (Nacalai Tesque, 06385−00) following the manufacture's protocol.

### Western blotting

Equal amount of proteins (10 μg) were resolved by SDS-PAGE using a gradient polyacrylamide gel (BIO CRAFT, SDG-571) and Running Buffer Solution for SDS-PAGE (Nacalai Tesque, 30329−61) and transferred to PVDF membranes (ATTO, WSE-4051). Membranes were blocked in 2% BSA in Tris buffered saline (TBS) buffer containing 0.1% Tween-20 (TBST) for 1 hour at room temperature. The blocked membrane was cut around 15 kDa region and the membranes of high and low molecular weight regions were incubated with Phospho-CK2 Substrate [(pS/pT)DXE] MultiMab® Rabbit mAb mix (Cell Signaling Technology, 8738, 1:1000 dilution) or Rabbit anti-histone H3 antibody (ProteinTech, 17168–1-AP, 1:1000 dilution) in TBST for 1 hour at room temperature, respectively. After washing with TBST two times, the membranes were incubated with Goat pAb to Rb IgG (HRP) (abcam, ab9751, 1:2000 dilution) in TBST for 30 min at room temperature. The protein bands were visualized using Chemi-Lumi One Super (Nacalai Tesque, 02230−14) and imaged with FUSION SOLO 2 (VILBER) system.

### Immunoprecipitation of bulk samples

10 μg of protein in 10 μL of RIPA buffer was fluorescently labeled with 20 μM SeTau-647-NHS (SETA BioMedicals, K9-4149) for 1 hour at room temperature. Antibody beads were prepared by incubating 300 μg of Dynabeads Protein G

(Invitrogen) with 0.5 µg of Phospho-CK2 Substrate [(pS/pT)DXE] MultiMab® Rabbit mAb mix (Cell Signaling Technology, 8738) or mouse IgG2b isotype control monoclonal antibody (Proteintech, 66360–3-Ig) in 5 µL HEPES buffered saline (HBS) buffer for 30 min at room temperature and washing the beads with HBS buffer once. The fluorescent-labelled proteins were mixed with the antibody beads in 20 µL of RIPA buffer and incubated for 1 hour at room temperature. After washing with HBS buffer containing 0.2% Tween-20 (HBST) three times, the beads were suspended with 10 µL of SDS-PAGE sample buffer and incubated at 90°C for 5 minutes. The immunoprecipitated samples were resolved by SDS-PAGE and imaged with FUSION SOLO 2 (VILBER) system.

## Single cell analysis of CK2 substrates phosphorylation

**Single-cell immunoprecipitation.** After the calyculin A treatment for 20 minutes, single HeLa cells gated using scattering light plots were sorted into 5 µL of RIPA buffer in BSA-passivated 96-well plates using FACS Aria II system (BD). Pseudo-single cell lysate was prepared by sorting 20 cells into 5 µL of RIPA buffer, making final volume to 100 µL with RIPA buffer, and distributing 5 µL of the diluted samples to each well of 96-well plate. As negative control, 5 µL of 50 ng/µL BSA solution in RIPA buffer was used instead of lysate. The lysates were incubated with 16.7 µM SeTau-647-NHS at room temperature for 1 hour. The antibody beads were prepared by incubating 150 µg of Dynabeads Protein G with 2 µg of antibodies for 1 hour and washing with HBS buffer once. Each cell lysate was mixed with 300 ng of antibody beads and incubated at 12°C overnight. After the incubation, the beads were washed with HBST buffer three times. To remove buffer efficiently while keeping bead pellets in wells, the plates were placed on magnet plates (BiT-Mag96, Sanplatec, BM-96) and all buffer removal was performed by centrifugation with the 96-well plates facing down. To embed the beads into a hydrogel, first the beads were incubated in 25 mM MgCl2 in 20 mM HEPES buffer supplemented with 0.1% Tween-20 for 3 min. After removing MgCl2 solution, the beads were incubated in 0.3% gellan gum in 20 mM HEPES buffer supplemented with 0.1% Tween-20 for 3 min so that magnesium ion retained in the bead pellets induce gelation of gellan gum. The gellan gum solution was removed by centrifugation and HBST buffer was added for pellet loading.

**Silane-coating of glass slides.** Silane solution was prepared by adding 40 ml of 3-(trimethoxysilyl)propyl methacrylate (Wako, 204−21205) and 60 ml of acetic acid (Nacalai Tesque, 00212−85) to 100 ml of milliQ water and degassed for 15 min in a sonicator. Plain glass microscope slides (Matsunami glass, S1112) were incubated in the silane solution for 30 minutes, washed with isopropyl alcohol (Nacalai Tesque, 29128−31) twice and milliQ water twice and dried with an air blow gun.

**Glass slide gel preparation.** A notched glass plate was designed using Fusion 360 (Autodesk) and made by OOKABE GLASS (S1a Fig). The plate was treated with Gel Slick Solution (Lonza, 50640) before gel preparation. A pair of spacers made from a plastic sheet with 0.1 mm thickness (Tamiya, 70208) were sandwiched between the silane-coated glass slide and the notched glass plate and fixed with clips (S1b Fig). The space between the glasses was filled with gel solution (6% acrylamide, 372 mM Tris-HCl (pH8.8), 0.1% SDS, 0.03% ammonium persurfate (APS), 0.08% *N,N,N',N'*-Tetramethylethylenediamine (TEMED)) and the assembly was incubated in a humidified plastic container at 40°C for 25–30 minutes. The unpolymerized solution on the gel top was removed with paper towel and an air blow gun to make space for bead pellet loading.

**Electrophoresis.** The bead pellets embedded in gellan gum gel were manually loaded using a P20 micropipette on the top of the glass slide gel on a custom-made magnetic stand (S2 Fig). The magnetic stand was designed in Fusion 360 (Autodesk, S2 File), 3D-printed using a Titan3 printer (MagnaRecta), and assembled with a neodymium magnet (NeoMag, NOS594). After loading, SDS-PAGE sample buffer (0.1% SDS and 373 mM Tris-HCl (pH8.8)) supplemented with 2% agarose heated to 70°C was added to the gel top. The glass slide gel was cooled in a fridge for more than 7 minutes to prevent the pellets moving in the following steps.

The electrophoresis chamber for glass slide gels was designed in Fusion 360 (Autodesk, S3 File) and 3D-printed using an ARM-10 printer (Roland) with imageCure resin (Roland, PRH35-ST). Stainless steel wire was used as the electrodes

and soldered to copper wires serving as electrical leads. A rubber sheet was attached to the surface of the upper chamber that comes into contact with the glass slide gel to prevent buffer leakage. The gel was run in 1x AllView PAGE Buffer (BioDynamics Laboratory Inc, DS520) at 50 V for 11 minutes.

**Single-molecule imaging using a PISA microscope.** After SDS-PAGE, the notched glass plate was removed and the gel on the glass microscope slide was incubated in 1 mM Trolox in PBS for 5 minutes at room temperature. The gel was placed on a fluorinated ethylene propylene (FEP) film (25 μm thick, Daikin Chemical, NF-0025) and set to the microscope stage.

A custom build microscope system (PISA) with a motorized stage (Prior Scientific, H117) controlled by an in-house program LabVIEW (National Instruments) was used to scan the gel for single-molecule imaging. The optical components and overall design of the PISA system have been described in detail previously (10). Briefly, PISA was built on a custom microscope body with two water-immersion objective lenses, fluorescence illumination (Special Optics, 54-10-7, NA = 0.66, 28.6×) and detection (Evident, XLUMPLFLN 20XW, NA = 1.0, 20×), that were placed below the coverslip at a tilted angle of 33.8 degrees. Bessel beam was generated by passing the laser source via an axicon lens (Mie Optics), which was then reflected by a Galvano mirror (Cambridge Technology, 6215HB), creating a light sheet. The detection port was connected to an EM-CCD camera (Andor, iXon Ultra 897) via an imaging lens and used to image fluorescence single molecules.

First, the bead pellets were manually located under bright field imaging. Next, each sample was scanned from the pellet over 1.6 cm to the direction of electrophoresis (y-axis) with a 10.73 μm step size. Because the PISA microscope illuminates and images optical slices at a tilted angle of 33.8°, the effective spacing between adjacent optical sections is approximately 6 μm. This spacing is comparable to the detectable thickness of the illuminated region and signal loss between adjacent optical sections is negligible [10]. Imaging was conducted using a 647 nm fiber laser (MPB Communications, 2RU-VFL-P-2000–647) at 800 mW and detected through a near-infrared band-pass filter (Semrock, FF01–708/75–25). The image size is 512 × 512 pixels, and one pixel corresponds to 0.8 μm. Images were saved in 16-bit TIFF format (pixel values range from 0 to 65,535) for further image analysis.

**Image analysis.** TIFF image sequences for each sample were processed using a custom Python script and the overview of the image analysis is shown in S4a Fig. For each frame, background subtraction was performed using a rolling-ball algorithm (radius = 25 pixels), followed by cropping to the region of the polyacrylamide gel (150 pixels in height and 312 or 412 pixels in width). The vertical crop position was adjusted linearly across frames according to start and end heights of the interface between the gel and glass slide signals to compensate for gradual sample movement.

Since the PISA microscope illuminates and images an optical slice at a tilted angle of 33.8°, deskewing is required to generate Z-axis projection images, which is computationally intensive. To save computational cost for the analysis, we generated 2D images projected to the tilted plane by shifting each cropped frame laterally by 11 pixels (≈ 8.8 μm) and computing the projection as the pixelwise maximum across all shifted frames (S4b Fig). The final projection images were saved as single TIFF files for each sample.

The positions of foci were determined using a Laplacian-of-Gaussian blob detector from the scikit-image package. The threshold for particle detection was set by manually inspecting overlays of detected particle markers on the projected images to ensure proper single-molecule detection. The same threshold was applied to the analysis of images obtained from the same gel scan. To avoid artifacts caused by impurities such as protein aggregates, bubbles, and dust, particles with signal intensity higher than 15,000 or with diameters larger than 2 pixels were removed (S4c Fig). The remaining particles were subjected to further analysis as signals derived from single protein molecules.

**PCA analysis.** The number of detected single molecules was binned at 20 pixels (16 μm). The counts were smoothed by calculating moving average of five consecutive bins and the smoothed data was subjected to principal component analysis (PCA) to calculate PCA scores and loading heatmaps.

## Supporting information

**S1 Fig. Assembly of the silane-coated glass slide and the notched glass plate.** (a) Design of the notched glass plate for glass slide gel preparation. (b) Assembly of the silane-coated glass slide and the notched glass plate.
(TIF)

**S2 Fig. Magnetic stand for bead pellet loading.**
(TIF)

**S3 Fig. Uncropped western blot and gel images.** Uncropped images of (a) the western blot shown in Fig 2a and (b) the gel shown in Fig 2b. The regions presented in the main figures are indicated by dashed boxes.
(TIF)

**S4 Fig. Workflow of the image analysis.** (a) Workflow of the image analysis pipeline used to convert a series of fluorescence images obtained from glass slide gel scanning into protein molecule profiling data. (b) Schematic illustration of a glass slide gel scan (left) and the projection of the acquired images (right). (c) Representative particle detection results with and without signal intensity and size filtering. Detected particles are indicated by yellow circles.
(TIF)

**S1 File. Representative projection images and their overlays with detected particles from the samples analyzed in Fig 2c.**
(ZIP)

**S2 File. 3D model data of the magnetic stand.**
(ZIP)

**S3 File. 3D model data of the electrophoresis chamber.**
(ZIP)

## Acknowledgments

We thank JCRB cell bank for providing HeLa cells (JCRB9004) and iCeMS Analysis Center for the support with single-cell sorting.

## Author contributions

**Conceptualization:** Takuya Hidaka.

**Funding acquisition:** Takuya Hidaka, Sooyeon Kim, Yuichi Taniguchi.

**Investigation:** Takuya Hidaka, Ryotaro Motoya, Gao Jintian, Sooyeon Kim.

**Supervision:** Yuichi Taniguchi.

**Writing – original draft:** Takuya Hidaka.

**Writing – review & editing:** Takuya Hidaka, Ryotaro Motoya, Gao Jintian, Sooyeon Kim, Yuichi Taniguchi.

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
