## [Decision Letter · Decision Letter 0]

23 Feb 2026

PONE-D-26-05037

Single-cell profiling of kinase substrate phosphorylation by single-molecule imaging

PLOS One

Dear Dr. Taniguchi,

We have received the comments from two expert reviewers who evaluated your work. Both reviewers find your work interesting and innovative and acknowledge the potential contribution of your study to the field. However, they have also raised a number of technical concerns that must be addressed before the manuscript can be accepted for publication.

We look forward to receiving your revised manuscript.

Kind regards,

Mauro Salvi, Ph.D.

Academic Editor

PLOS One

**Journal Requirements:**

“This work was supported by Japan Society for the Promotion of Science (JSPS) [22K14786 (Grant-in-Aid for Early-Career Scientists) and 22KJ3130 (Grant-in-Aid for JSPS Fellows) to T. H., 20H00460 (Grants-in-aid for Scientific Research (A)), 19H05545 and 20K20458 (Grants-in-aid for Challenging Pioneering Research) to Y. T., 19K15718 and 22K14800 (Grants-in-aid for Early-Career Scientists) to S. K.], Japan Science and Technology Agency [JPMJPR25J4 (PRESTO) and JPMJAX1914 (ACT-X) to S. K. and JPMJCR2334 (CREST) to Y. T.], Suntory Foundation for Life Science [SunRiSE] to Y. T., and The Asahi Glass Foundation [Continuation Grants for Young Researchers] to Y. T.”

5. We note that Figures 1, Fig S1, and Fig S2 in your submission contain copyrighted images. All PLOS content is published under the Creative Commons Attribution License (CC BY 4.0), which means that the manuscript, images, and Supporting Information files will be freely available online, and any third party is permitted to access, download, copy, distribute, and use these materials in any way, even commercially, with proper attribution. For more information, see our copyright guidelines: http://journals.plos.org/plosone/s/licenses-and-copyright.

1. You may seek permission from the original copyright holder of Figure(s) [#] to publish the content specifically under the CC BY 4.0 license.

**Additional Editor Comments:**

Dear Authors, we have received the comments from two expert reviewers who evaluated your work.

Both reviewers find your work interesting and innovative and acknowledge the potential contribution of your study to the field.

However, they have also raised a number of technical concerns that must be addressed before the manuscript can be accepted for publication.

Reviewers' comments:

Reviewer's Responses to Questions

**Comments to the Author**

1. Is the manuscript technically sound, and do the data support the conclusions?

Reviewer #1: Yes

Reviewer #2: Partly

2. Has the statistical analysis been performed appropriately and rigorously?

Reviewer #1: N/A

Reviewer #2: No

3. Have the authors made all data underlying the findings in their manuscript fully available?

Reviewer #1: Yes

Reviewer #2: Yes

4. Is the manuscript presented in an intelligible fashion and written in standard English?

Reviewer #1: Yes

Reviewer #2: Yes

5. Review Comments to the Author

Reviewer #1: In this manuscript, authors address the challenge of analyzing low abundant phosphoproteins at single-cell level. Building upon their previous studies on the PISA microscope and single-cell PAGE-PISA (referenced on bioRxiv [10, 11]), they present a methodological approach to profile kinase substrate phosphorylation in individual cells. The workflow involves fluorescence labeling of the proteome, enrichment of phosphorylated substrates containing specific amino acid motifs via immunoprecipitation using motif-recognizing antibodies, and detection of single molecules through gel electrophoresis coupled with a custom-built imaging system. Using HeLa cells, the authors analyze CK2 substrates and demonstrate that their method enables single-cell-level comparison of phosphorylation patterns, with phosphoproteins separated by molecular weight prior to single-molecule detection.

Comments:

1. Reference papers for foundational methodological work: Given that the foundational methodological works [10, 11] are available as preprints, clarification regarding their current publication status would help contextualize the robustness and maturity of the approach described here.

2. Single-cell variability and experimental error control: While the included specificity controls (e.g., isotype control immunoprecipitation and BSA immunoprecipitation) support the selectivity of the assay, the manuscript would benefit from a clearer explanation of how experimental variability at the single-cell level was controlled. If internal references (e.g., spike-in controls or other normalization strategies) were used to monitor technical fluctuations between samples, please clarify. Otherwise, a brief discussion of how technical variability was distinguished from biological heterogeneity would further strengthen the robustness of the conclusions.

3. Figure 1C - labeling: For clarity, please specify in the figure legend what the labels “1” and “2” represent in Figure 1C (e.g., initial bright-field imaging (1) followed by fluorescence image scanning (2)).

4. Figure 1D - scale bar: It would improve clarity to include an additional scale bar in the zoomed panel of Figure 1D.

5. Statistical analysis (Figure 2E): While the visual comparison in Figure 2E indicates greater variability in single-cell samples, providing a statistical evaluation of variability would strengthen the interpretation.

6. Figure organization (Figure S1): Figure S1 provides methodological clarification that appears conceptually earlier in the experimental workflow than Figures S2 and S3. However, it is currently cited later in the text. Adjusting the placement of Figure S1 (e.g., integrating into Figure 1B or revising the citation order) may improve the logical flow of the manuscript.

7. Reproducibility and accessibility of gel electrophoresis setup (Figure S2): Additional details regarding the fabrication and availability of the gel prepation and electrophoresis setup (e.g., the 3D-printed gel casts) would improve reproducibility. Are the design files publicly available, or can they be requested from the authors?

8. Microscopy step size and potential signal loss: The rationale for selecting the microscopy step size should be clarified. Specifically, how was it determined relative to the expected protein size or diffusion characteristics? Providing clarification on whether signal loss between optical sections, particularly for smaller proteins, is expected to be negligible would further strengthen confidence in the detection strategy.

Overall, the study presents a technically sound and innovative approach. The suggested revisions primarily concern clarification and statistical strengthening rather than fundamental methodological concerns.

Reviewer #2: General Assessment

This manuscript describes a novel and creative method for single-cell phosphoprotein profiling integrating fluorescence labeling, motif-specific immunoprecipitation, miniaturized SDS-PAGE, and single-molecule imaging using PISA microscopy. The approach is technically innovative and enables visualization of cell-to-cell heterogeneity in CK2 substrate phosphorylation. The system appears highly sensitive and could represent a useful platform for studying kinase signaling at single-cell resolution. The authors’ discussion of potential future coupling with mass spectrometry to enable protein identification further strengthens the conceptual scope of the work.

However, in its current form, the method does not permit identification of the proteins contributing to each detected peak, which substantially limits biological interpretation. The standalone utility of the method therefore requires clearer articulation. In addition, several methodological, reproducibility, and interpretational issues should be addressed prior to publication in PLOS ONE.

Major Concerns

1. Protein Identification and Scope of Interpretation

• Although the authors discuss potential coupling with mass spectrometry, the current study does not allow identification of the proteins underlying each molecular weight region. This limitation should be more explicitly stated and discussed in the manuscript.

• The method relies on immunoprecipitation using an antibody against the (pS/pT)DXE motif. This approach captures only a subset of CK2 substrates and may also pull down substrates of other kinases sharing this motif. The manuscript should clearly acknowledge:

o That only motif-containing substrates are assessed.

o That the detected proteins cannot be definitively assigned as CK2 substrates without orthogonal validation.

o That substrates lacking this motif are not captured by the current system.

Greater clarity here would prevent overinterpretation of the data.

2. BSA-like Peak in Isotype Control (Figure 2C)

A peak corresponding to the migration position of BSA appears in the isotype control condition in Figure 2C. The authors should clarify:

• The origin of this peak.

• Whether it reflects residual nonspecific binding, antibody contamination, or background labeling.

• Why this peak appears in both treated and untreated samples.

This is important for assessing background levels and specificity.

3. Pseudo-Single-Cell vs. Single-Cell Heterogeneity (Figures 2E and 2F)

In Figure 2E, one treated single-cell replicate appears to be a potential outlier. Because the heterogeneity analysis and PCA in Figure 2F heavily rely on 2E dataset, an outlier could disproportionately influence conclusions.

The authors should:

• Confirm whether this sample reflects biological variability or technical artifact.

• Consider repeating the experiment or increasing replicate number.

• Re-evaluate PCA and heterogeneity analysis after addressing this issue.

Given that the primary conclusion of the manuscript is the detection of cell-to-cell heterogeneity, robustness of this analysis is essential.

4. Concordance Between Profiles in Figures 2C, 2D, and 2E

When comparing profiles across Figures 2C, 2D, and 2E, the migration patterns and signal distributions do not appear fully concordant.

The authors should comment on:

• The reproducibility of the profiling method across experiments.

• Sources of variability between runs (e.g., gel preparation, labeling efficiency, IP efficiency).

• Quantitative metrics of reproducibility, if available.

Clarifying inter-experimental variability would strengthen confidence in the method.

5. Fluorophore-to-Protein Stoichiometry and Quantitative Interpretation

The method relies on NHS-ester labeling, which depends on lysine number and accessibility. This raises important quantitative considerations:

• How do the authors account for variability in labeling stoichiometry across different proteins?

• Is labeling assumed to be proportional to protein abundance?

• How might differential lysine content affect comparison across molecular weight regions?

Additionally, when multiple proteins of similar molecular weight co-migrate, how is signal overlap addressed? These issues should be discussed explicitly, even if only as methodological limitations.

Minor Concerns

1. Antibody Specificity

• The anti-(pS/pT)DXE antibody recognizes a motif shared among multiple proteins and is not exclusive to CK2 substrates. The manuscript should clarify that “CK2 substrates” are inferred based on motif recognition rather than direct kinase validation.

• The limitation that not all CK2 substrates contain this motif, and that other kinases may generate similar motifs, should be clearly stated.

2. Materials and Methods – Immunoprecipitation of bulk samples

• Please list all antibodies used with Protein G beads (source and catalog number).

• The BSA control experiment shown in Figure 2C should be described in the Methods section.

3. Materials and Methods – Image Analysis

• The criteria used for thresholding and distinguishing true single-molecule foci from aggregates, bubbles, or dust should be described in greater detail.

• Inclusion of representative raw images (including excluded artifacts) in the supplementary material would improve transparency.

Summary

This manuscript presents a technically innovative platform for single-cell phosphoprotein profiling. The method appears sensitive and capable of detecting heterogeneity at the motif level, and the discussion of future integration with mass spectrometry enhances its conceptual appeal.

However, clarification of key methodological limitations, quantitative assumptions, antibody specificity, reproducibility across experiments, and interpretation of the heterogeneity analysis is necessary. Addressing these issues would significantly strengthen the manuscript and make it suitable for publication in PLOS ONE.

6. PLOS authors have the option to publish the peer review history of their article (what does this mean?). If published, this will include your full peer review and any attached files.

**Do you want your identity to be public for this peer review?** For information about this choice, including consent withdrawal, please see our Privacy Policy.

Reviewer #1: No

Reviewer #2: No

---

## [Author Response · Author response to Decision Letter 1]

13 Apr 2026

Please see the uploaded "Response to Reviewers" file (Response_to_Reviewers.pdf) for our detailed point-by-point response to all reviewer comments, including figures.

---

## [Decision Letter · Decision Letter 1]

12 May 2026

PONE-D-26-05037R1Single-cell profiling of kinase substrate phosphorylation by single-molecule imagingPLOS One

Dear Dr. Taniguchi,

Thank you for submitting your manuscript to PLOS ONE. After careful consideration, we feel that it has merit but does not fully meet PLOS ONE’s publication criteria as it currently stands. Therefore, we invite you to submit a revised version of the manuscript that addresses the points raised during the review process.

The revised manuscript has improved. Please solve the minor issues raised by Reviewer 1.

We look forward to receiving your revised manuscript.

Kind regards,

Mauro Salvi, Ph.D.

Academic Editor

PLOS One

Journal Requirements:

Reviewers' comments:

Reviewer's Responses to Questions

**Comments to the Author**

1. If the authors have adequately addressed your comments raised in a previous round of review and you feel that this manuscript is now acceptable for publication, you may indicate that here to bypass the “Comments to the Author” section, enter your conflict of interest statement in the “Confidential to Editor” section, and submit your "Accept" recommendation.

Reviewer #1: (No Response)

Reviewer #2: All comments have been addressed

2. Is the manuscript technically sound, and do the data support the conclusions?

Reviewer #1: Yes

Reviewer #2: Yes

3. Has the statistical analysis been performed appropriately and rigorously? 

Reviewer #1: N/A

Reviewer #2: Yes

4. Have the authors made all data underlying the findings in their manuscript fully available?

Reviewer #1: Yes

Reviewer #2: Yes

5. Is the manuscript presented in an intelligible fashion and written in standard English?

Reviewer #1: Yes

Reviewer #2: Yes

6. Review Comments to the Author

Reviewer #1: I thank the authors for their detailed responses and the clarifications provided. The revised manuscript successfully addresses most of my previous concerns. However, there are a few remaining points that should be addressed before final publication:

1. Methodological Solidity: I note that reference [10] remains a preprint and has not yet been submitted for peer review, and reference [11] is also not yet formally published. As these works underpin key methodological aspects, their current status limit how solidly the present study can be assessed. I recommend the authors update these citations if they have since been accepted, or provide further justification for relying on unpublished work for core methods.

2. Quantitative Validation of Variability: While Figure 2F visually indicates a shift in standard deviation, the manuscript still lacks a formal quantitative/statistical evaluation to confirm this difference. Briefly acknowledging in the Discussion that establishing robust quantitative metrics remains a challenge (as noted in the rebuttal) would provide a more transparent context for the results.

3. Figure Quality: The figures provided require higher resolution to ensure clear representation and readability in the final layout.

4. Typological Errors: There are formatting issues in lines 69, 93, 96, and 444, where sentences appear to end prematurely before the referenced citations. Please review the manuscript for consistent punctuation and citation placement.

Overall, the current version presents a potentially sound methodological approach, provided that the supporting references are formalized, and it correctly addresses the study's limitations. Once these minor technical and formatting issues are resolved, the manuscript will be in a much stronger position for publication.

Reviewer #2: (No Response)

7. PLOS authors have the option to publish the peer review history of their article (what does this mean?). If published, this will include your full peer review and any attached files.

Reviewer #1: No

Reviewer #2: No

---

## [Author Response · Author response to Decision Letter 2]

14 May 2026

The Response to Reviewers letter has been uploaded.

---

## [Editor Report · Decision Letter 2]

17 May 2026

Single-cell profiling of kinase substrate phosphorylation by single-molecule imaging

PONE-D-26-05037R2

Dear Dr. Taniguchi,

We’re pleased to inform you that your manuscript has been judged scientifically suitable for publication and will be formally accepted for publication once it meets all outstanding technical requirements.

Kind regards,

Mauro Salvi, Ph.D.

Academic Editor

PLOS One
---

## [Editor Report · Acceptance letter]

PONE-D-26-05037R2

PLOS One

Dear Dr. Taniguchi,

I'm pleased to inform you that your manuscript has been deemed suitable for publication in PLOS One. Congratulations! Your manuscript is now being handed over to our production team.

Kind regards,

on behalf of

Prof. Mauro Salvi

Academic Editor

PLOS One